# The Need for Multi-Omics Biomarker Signatures in Precision Medicine

**DOI:** 10.3390/ijms20194781

**Published:** 2019-09-26

**Authors:** Michael Olivier, Reto Asmis, Gregory A. Hawkins, Timothy D. Howard, Laura A. Cox

**Affiliations:** 1Center for Precision Medicine, Department of Internal Medicine, Wake Forest Baptist Health Comprehensive Cancer Center, Wake Forest University Health Sciences, Winston-Salem, NC 27157, USA; rasmis@wakehealth.edu (R.A.); laurcox@wakehealth.edu (L.A.C.); 2Center for Precision Medicine, Department of Biochemistry, Wake Forest Baptist Health Comprehensive Cancer Center, Wake Forest University Health Sciences, Winston-Salem, NC 27157, USA; ghawkins@wakehealth.edu; 3Center for Precision Medicine, Department of Biochemistry, Wake Forest University Health Sciences, Winston-Salem, NC 27157, USA; tdhoward@wakehealth.edu

**Keywords:** genomics, proteomics, metabolomics, transcriptomics, epigenomics, integrated multi-omics, precision medicine, precision oncology

## Abstract

Recent advances in omics technologies have led to unprecedented efforts characterizing the molecular changes that underlie the development and progression of a wide array of complex human diseases, including cancer. As a result, multi-omics analyses—which take advantage of these technologies in genomics, transcriptomics, epigenomics, proteomics, metabolomics, and other omics areas—have been proposed and heralded as the key to advancing precision medicine in the clinic. In the field of precision oncology, genomics approaches, and, more recently, other omics analyses have helped reveal several key mechanisms in cancer development, treatment resistance, and recurrence risk, and several of these findings have been implemented in clinical oncology to help guide treatment decisions. However, truly integrated multi-omics analyses have not been applied widely, preventing further advances in precision medicine. Additional efforts are needed to develop the analytical infrastructure necessary to generate, analyze, and annotate multi-omics data effectively to inform precision medicine-based decision-making.

## 1. Introduction

A major goal of biomedical research is to identify accurate, early indicators of disease. Over the past decades, advances in technology have ushered in an unprecedented era in biomedical research. Rather than focusing research efforts on individual molecules, pathways, or cells of interest, new technologies now allow characterization of complex biological systems in great detail and at unparalleled resolution. Sequencing technologies have helped decipher the genetic code of numerous organisms, including humans. Imaging and microscopy technologies have revolutionized our ability to visualize and monitor cellular and molecular processes in cells and tissues, and other advances in analytical methodologies such as NMR, mass spectrometry, and array-based and microfluidics methodologies all have contributed to a rapidly growing knowledgebase about the molecular composition and function of biological systems. This knowledge, in turn, has formed the basis for our continuing research efforts to better understand and characterize the molecular and cellular mechanisms contributing to the development and progression of diseases in humans and model organisms. The general promise and primary goal of precision medicine is to identify more accurate, earlier indicators of health trajectories for individuals, the detection of early stages of disease development, revert disease development, slow disease progression, and alter health trajectories through targeted and more effective pharmacological treatments or lifestyle interventions.

Unfortunately, the implementation of precision medicine approaches, whether in oncology or other fields of clinical medicine, has not kept pace with our ability to generate large-scale molecular data and information. In fact, our ability to analyze and interpret the data for translation into clinically actionable information has been challenging, and the initial promise of genomic and other molecular data as key contributors to clinical decision making has been slow to translate to clinical practice. With rapid technical advances and opportunities for more cost-efficient large-scale data generation in biomedical research, it is likely that our main challenge will continue to be the development of methods for extracting useful information from these complex, multidimensional datasets to guide clinical practice.

The technological advances mentioned above have created various new fields of research, commonly referred to as “omics”. Beginning with the field of genomics, the development of novel sequencing technologies now allows the cost-effective and rapid elucidation of an entire genome and the study of all genes at once, rather than gene by gene analysis. The term “genomics” highlights this comprehensive analysis [1]. A whole range of additional omics technologies has been developed, with ‘omics’ referring to the comprehensive study of the roles, relationships, and actions of various types of molecules in cells of an organism. This includes fields such as transcriptomics (the study of the expression of all genes in a cell or organism [2,3,4], proteomics (the analysis of all proteins, [5,6]), metabolomics (the comprehensive analysis of all small molecules, [7,8]), and epigenomics (the study of the epigenetic regulation of the entire genome, [9,10]), to name a few. It has also yielded a plethora of other ‘omics’ terms, such as lipidomics [11,12], metagenomics [13], glycomics [14], connectomics [15], cellomics [16], and even foodomics [17,18]. While a detailed discussion of these omics technologies exceeds the focus of this review, we will first discuss applications of individual omics technologies to study cancer and other human disorders (summarized in Figure 1), and then provide examples of the initial steps of ‘multi-omics’, the integrative use of multiple omics platforms. Several comprehensive reviews have also discussed the use of multiple omics platforms in cancer research, and provide an in-depth overview about the current state of the field (e.g., [19]). Here, we will focus our discussion on representative examples of integrated omics studies, and then outline the challenges and opportunities for this emerging field of multi-omics for precision medicine and oncology.

## 2. Genomics Approaches

Most often, a precision medicine approach is equated with a genomics approach. This is likely due to the fact that genomic approaches, technologies, and data were the first widespread omics data available for application in precision medicine. As a result, most early efforts in precision medicine predominantly focused on the use of genomic sequence information to diagnose patients, predict risk of individual patients for developing diseases, and to assess whether specific treatments are suitable and likely successful in individual patients. Over the past decade, genome-wide association studies, and more recently, next generation whole exome and whole genome sequencing data have provided a large set of DNA sequence variants that are associated with a plethora of diseases and traits in humans. Despite these rapidly expanding datasets, progress in identifying the specific disease-causing sequence variants and the pathophysiological mechanisms by which they affect disease development or progression has been slow. The application of polygenic risk score approaches to a range of common diseases has been one strategy to provide potentially clinically useful and actionable information from these genetic data without the need to identify individual causal sequence changes and their modes of action. However, application of these risk scores has been limited to only a few diseases. The extensive genetic variant information has also been used in pharmacogenomics to predict drug responses in individual patients. Similar to risk scores, this strategy has had limited success. Clearly, more progress is needed to effectively use available genomic information and assess benefits and limitations in risk prediction, disease diagnosis, and treatment decision making.

In oncology, genomics approaches have focused on DNA sequencing to identify cancer-specific mutations, both in heritable and spontaneous cancers, and to analyze chromosomal rearrangements to characterize cancer classes or subtypes. DNA isolated from cancer cells carries a wide range of somatic and potentially inherited changes, ranging from single nucleotide polymorphisms, insertions and deletions and larger copy number variations and structural rearrangements. The total number of mutations and/or chromosomal rearrangements varies widely between tumor types, and between the same tumor types from different individuals, making it challenging to identify key functional changes that are essential for tumor formation and growth. The overall mutational pattern in human cancer has been extensively studied over the past decade and several comprehensive reviews have summarized the results [20,21,22].

Initial genetic oncology studies focused on the identification of inherited mutations that significantly elevate cancer risk. Early studies of families with a clustering of particular cancer types led to the identification of a number of genes and mutations that, when inherited, increase the risk for specific types of cancer. Initial examples include Lynch Syndrome, a non-polyposis colorectal cancer often caused by mutations in genes involved in DNA mismatch repair (MLH1, MSH2, MSH6, PMS1, PMS2, [23]), Li-Fraumeni Syndrome, an inherited syndrome most often caused by mutations in the tumor suppressor gene TP53 that can lead to the development of a wide range of cancers, including sarcomas, leukemia, and brain cancers [24], and hereditary breast and ovarian cancer syndrome, often caused by mutations in the genes BRCA1 and BRCA2 [25]. Individuals with mutations in any of these genes have a significantly higher risk of developing cancer compared to non-carriers. These discoveries facilitated the use genetic information for treatment decisions in affected individuals and have led to a wide range of providers that offer genetic testing for any of these conditions—a first application of genetics/genomics in precision oncology. As an extension of these early studies, genome-wide association studies [26] and, more recently, the development of polygenic risk scores [27] have expanded the number of sequence variants and mutations that are associated with increased cancer risk. A significant number of these variants are routinely tested today as companion diagnostics for tumor characterization and staging, and treatment decision-making.

Additional studies have focused on the analysis of chromosomal rearrangements and copy number variation in cancer cells. Numerous technologies, including fluorescence in-situ hybridization (for karyotypic characterization), comparative genomic hybridization (CGH, for small chromosomal rearrangements and copy number variation), and recently next generation whole genome sequencing, allow the detailed characterization of chromosomal abnormalities predisposing to cancer development, and the identification cancer cell-specific alterations that help explain the abnormal cell growth or resistance to treatment, as in the case of the formation of fusion genes such as BCR-ABL (Philadelphia Chromosome) [28]. Lately, next generation whole genome sequencing has also been used to characterize gene conversion by inter-chromosomal recombination by comparing tumor tissues for several cancers with adjacent “paratumor” tissues [29].

Despite this tremendous progress in clinical testing for inherited forms of cancer, progress towards better understanding the underlying genetic mechanisms contributing to the majority of sporadic cancers has been more challenging. Recent advances in sequencing technologies now allow the effective analysis of individual tumors, including sequence analysis at the single cell level. The growing set of data available from these efforts have revealed that the underlying profile of somatic mutations accumulating in tumor cells is highly complex, and it has been challenging to distinguish potential ‘driver’ mutations (that are responsible for tumor formation and growth) from ‘passenger’ mutations that are phenotypically neutral. The understanding of mutational burden in tumors, however, has been used to better define the pathophysiological mechanisms contributing to tumor development and progression. Since 2005, The Cancer Genome Atlas (TCGA), an NIH-funded effort to characterize and profile the genome of 33 different tumor types and 10 rare cancers, sequenced the genomes of over 7500 samples to identify sequence variants [30], other DNA alterations such as fusions [31,32], copy number alterations [33,34], and other complex structural variations [35]. The large dataset has already resulted in more accurate stratification and prognosis of cancer subtypes [36,37,38], and identification of molecular subtypes of cancer that can be treated by available drugs [39,40], highlighting the steady progress made in using genetic information in clinical oncology and practice. Nonetheless, despite these success stories, it has become clear that an exclusive characterization of the genomic state of a tumor cell, or a characterization of the genome of an individual, is often not sufficiently predictive of their risk of developing cancer, nor of their likelihood to respond to treatment or their cancer recurrence risk.

## 3. Other Omics Approaches

To better characterize tumor cells and their functional abnormalities, other omics technologies have been used across the entire spectrum of human disorders. Virtually all of these approaches were also applied to tumor and cancer samples in the past several years and are beginning to contribute to a better understanding of the molecular mechanisms underlying disease processes. Similar to the chapter above summarizing the genomic approaches and technologies applied in cancer research, we will not attempt to provide a comprehensive review of all omics technologies and approaches that have been applied over the years, and how they individually have impacted and altered clinical practice. Rather, we will highlight a number of applications and discuss the resultant clinical use of the data and information, before we discuss the prevailing challenges in using omics data effectively and comprehensively in the next chapter.

### 3.1. Transcriptomics

As sequencing and array-based technologies advanced, disease genetics research has extensively used these approaches to profile and quantitatively analyze the transcriptome of cells and tissues, and determine how changes in gene transcription in cells can be used to diagnose disorders and identify molecular mechanisms underlying the disease development and progression. These transcriptome data have also spawned an entire new approach, expression quantitative trait loci (eQTL) analysis, to identify putative functional mechanisms by which genetic sequence variation may lead to a disease by linking DNA sequence variation with changes in gene expression.

To date, almost 800,000 gene expression datasets related to cancer have been deposited in the Gene Expression Omnibus (GEO) database of the National Center for Biotechnology (NCBI) of the NIH. The TCGA initiative has also used transcriptomics to provide detailed gene expression characterization of individual tumors and tissues from over 11,000 patients [41], and together with other initiatives such as the Stand Up To Cancer-Prostate Center Foundation Project [42], has helped in the detailed analysis of gene expression changes associated with tumor formation and growth. Numerous other studies have also revealed characteristic gene expression signatures that can help predict patient outcomes and treatment response for a variety of cancer types, and several clinical tests are now on the market that predict prognosis or recurrence risk for patients with breast cancer, colorectal cancer, glioblastoma multiforme, and non-small lung cancer [43,44,45,46]. Recent studies have extended gene expression analysis to single cells [47], and the resulting data are likely to further expand our understanding of cell heterogeneity in cancer [48], and may influence our clinical decision making.

### 3.2. Epigenomics

With a growing understanding of gene expression changes in tissues and cells, research has extensively focused on studying the regulation of gene expression. In cancer, this regulation is highly complex, with germline genetic factors, somatic mutations, and epigenetic contributors all affecting gene expression [49]. This interplay has been extensively studied in various cancers, including eQTL analysis to identify inherited genetic factors contributing to gene expression variation (e.g., [50,51]), and single cell gene expression analysis that allows the assessment of the impact of cell-specific somatic mutations on gene expression in individual cells [52,53,54]. In addition to these studies, epigenetic regulatory mechanisms play a key role in cancer cell formation and growth. This includes alterations in DNA methylation [55], histone modifications [56], and expression of miRNAs [57] and lncRNAs [58,59]. Epigenetic differences can be observed between tumor cells and normal cells, and in tumor cells during disease progression or in response to treatment (mostly chemotherapy). Accordingly, DNA methylation has been used to prioritize treatment regimens for some conditions. DNA methylation status at the MGMT promoter is currently used to determine if temozolamide, a DNA alkylating agent, will be effective in glioblastoma treatment [60]. MGMT is a DNA repair enzyme, and when the gene’s promoter is methylated, is expressed at low levels. Treatment with temozolamide is effective in this case because the damage to cancer cells induced by the treatment cannot be repaired, leading to cell death. Conversely, when the MGMT promoter is not methylated, the gene is expressed and the damaging effect of temozolamide is limited by the cell’s ability to rescue the damage [61]. Altered expression of miRNAs has been observed in lung cancers that have become resistant to treatment with doxorubicin [61], and other epigenetic changes have been observed in other cancer types. As a result, DNA methyltransferases or histone deacetylase inhibitors that may interfere with these mechanisms have been tested as putative cancer drugs, and have shown promising preclinical results in pancreatic cancer [62]. Similarly, expression profiles of specific miRNA and lncRNA molecules may help in the identification of drug-resistant tumor cells in patients undergoing standard chemotherapy, identifying them as patients in need of additional treatment. Ultimately, miRNAs and lncRNAs may provide opportunities for novel treatment approaches in cancers that are not effectively targeted by currently available treatment options, such as glioblastoma [63]. However, it is unlikely that epigenomic signatures alone, or treatments aimed at interfering with epigenetic mechanisms in cancer, will be successful for treatment of all types of cancer.

### 3.3. Proteomics

As gene expression analysis has advanced, investigations have turned to the analysis of cellular proteins, the translational ‘products’ of RNA transcripts, and the predominant mediators of cellular function. While transcriptomics allows the quantification of the immediate product of a cell’s genome at a particular time (thus ‘measuring’ the direct activity of the genome, or any change thereof under different conditions or during disease development), initial large-scale studies of cellular proteomes showed a relatively low correlation between protein expression levels and corresponding mRNA expression levels [64]. Clearly, protein function is mediated and altered by a myriad of mechanisms. Posttranslational modifications of proteins (e.g., phosphorylation) often are required for activity or signaling. Folding and posttranslational processing of pre-proteins, and the formation of multi-protein complexes are necessary to create the machinery that preforms the required cellular functions. Finally, cellular location of a protein often determines whether an expressed protein is active or inactive. As a result, no single methodology is able to appropriately assess the ‘proteome’ in all its facets, and most proteomics methodologies focus on the identification and quantification of individual proteins, usually by mass spectrometry or through affinity-based protein arrays. Additional analytical methodologies, such as NMR and X-ray crystallography, can provide information on the structure of proteins and protein complexes in cells and tissues.

In cancer proteomics, the TCGA Consortium represented the first large-scale effort to profile the tumor proteome. However, the analysis was performed using reverse phase protein arrays, and was therefore limited to the targeted analysis of a few hundred proteins. However, several recent studies have used state-of-the-art mass spectrometry approaches to identify cancer-specific biomarkers in ovarian cancer [65], and used proteomics data to aid in the classification of breast cancer [66]. Additional analyses have also used proteomics data to predict drug sensitivity and identify putative proteins mediating drug resistance [67,68]. These proteome-wide studies complement the traditional immunohistochemical classification of tumor types, such as the characterization of estrogen receptor expression in breast cancer tumors.

### 3.4. Metabolomics

Metabolomics, the analysis of small molecules present in a cell, tissue, or fluid, has been the focus of biomarker discovery studies for a long time. Metabolites are often viewed as the products of cellular processes, mediated by proteins; therefore, changes in metabolites are presumed to be reflective of changes in function of the mediating enzymes and proteins. The vast majority of metabolomics analyses have focused on the analysis of plasma or serum samples from patients, often as an attempt to identify cancer or tumor-specific biomarkers that can be used in diagnosis without requiring an invasive tumor biopsy sample. However, as in most disease-related studies, determining the specificity of a characteristic change in plasma metabolites for a particular disease, or, in the case of cancer, tumor type, remains difficult, and often metabolites identified as biomarkers in one disease are also altered in other diseases.

The limited studies in cancer biology have identified some putative metabolic biomarkers, such as altered carbohydrates in acute myeloid leukemia [69], and unsaturated free fatty acids in colorectal cancer [70]. In contrast, studies in prostate cancer only revealed changes in citrate [71] and branched-chain amino acids [72], which have been reported for other diseases [73,74], suggesting that the observed biomarkers are likely not specific to prostate cancer and more likely reflective of shared underlying disease pathology mechanisms. Furthermore, it remains to be seen whether changes identified in plasma metabolites are representative of changes in tumor cells.

## 4. Multi-Omics Approaches: Challenges and Opportunities

As highlighted above, different omics datasets do not overlap extensively, and measures obtained from one omics approach are often not well correlated with data obtained by other methods. Thus, it is likely that different omics methodologies assess different parts of the complex pathophysiology of complex disease development and progression, and the analysis of just one omics subset provides a skewed, biased, and incomplete picture of the underlying biology. However, given the wide range of data that can be generated from tissue samples to characterize differences between normal and diseased cells and tissues, how does one select the most meaningful omic data types to generate (limited primarily by cost and tissue availability), and, more importantly, how does one integrate the resulting multiple omic datasets to obtain a comprehensive picture of the underlying biological processes? Essentially, there are two potential approaches to investigate multi-omics data:

The first approach looks at the various analytes (transcripts, proteins, metabolites, epigenetic factors) in the context of known (reported) pathways and mechanisms. This approach requires prior knowledge that specific molecular pathways are central to the disease process, and a detailed knowledge and annotation of all relevant molecular pathways, with annotations and identifiers that easily link back to transcripts, proteins, metabolites, miRNAs, lncRNAs, genetic sequence variants, and DNA and histone methylation variants, to list just a few. Recent consortium efforts, such as the Encyclopedia of DNA Elements (ENCODE) have begun to link gene expression regulation to DNA sequence variation, epigenetic variation, and chromatin accessibility on a genome-wide scale in various cell types, including cancer cell lines. While this is highly informative, it remains unclear how representative this information is for other cells and tissues, such as specific tumor tissues only distantly related to the cell lines under study. Other omics data can potentially be linked using gene names/protein names, as they have been annotated in databases such as KEGG and other pathway-driven annotations. However, the further data integration becomes complicated since the ENCODE- and other regulatory data do not necessarily directly link to individual genes which in turn can be mapped to KEGG pathways. Likewise, metabolite data are only sparsely linked in the current versions of gene-centric pathway networks. This brief illustration highlights the challenges of true multi-omics data integration, and is likely the reason why most manuscripts referring to multi-omics analyses only analyze data from two or three different platforms (e.g., transcripts and proteins).

The second approach ignores (at least initially) the existing knowledge of pathways and network interactions in cells and tissues, and looks for correlations across multiple data sets to identify molecules/analytes that are changing in a correlated manner. In principle, such an approach would be agnostic to the type of omics data used. The benefit of this approach is ability to discover novel molecules and pathways essential for the disease process. However, such an unbiased analysis very quickly becomes computationally and statistically challenging. The dataset may include RNA-Seq data on 20,000 different transcripts, but proteomics data only on 2,000 proteins. Furthermore, some data are generated using platforms that exhaustively analyze and quantify the molecules of interest (e.g., RNA-Seq), some may be unbiased, but only identify and quantify the most abundant molecules (e.g., proteomics), and yet other methodologies may use targeted approaches to investigate samples (such as targeted metabolomics approaches, array-based DNA methylation platforms). Finally, a complete multi-omics dataset may include over 100,000 analytes, yet they are likely to be measured in only a limited number of tissue samples (100 tumor biopsies), or potentially even in different samples from the same patient (tumor RNA-Seq vs. plasma metabolomics analysis). Analytical tools have been suggested for these types of complex datasets, such as clustering approaches [75], or weighted gene co-expression network analysis (WGCNA, [76]). However, most of these have not been applied extensively to datasets including more than two or three omics datasets. Additional tools will be needed to perform these complex integrated analyses using strict statistical models, and to help in the interpretation and biological annotation of the obtained co-expression networks or clusters of multi-omics analytes [77].

Numerous recent reviews and opinion pieces have suggested that the field of precision oncology will greatly benefit from a multi-omics analytical approach [19,78,79]. Indeed, this suggestion is not specific to the field of cancer research, and other human diseases would likely benefit from similar approaches [80]. Repeatedly, the argument is being made that only the comprehensive integrated analysis will be able to uncover the complex mechanisms underlying cancer development and progression. Despite this push, there are only limited examples of multi-omics studies. In fact, even the TCGA efforts only explored and integrated a limited number of omics approaches, and even though several manuscripts describe so-called multi-omics analyses of various cancer types, most of these analyses integrated only transcriptome and limited epigenome analysis with tumor mutation analysis, and a sparse targeted proteomics dataset. Similarly, virtually all analyses published to date focus primarily on transcriptomics, and add a limited amount of data from additional technologies to complement the RNA expression data, either as a proteogenomics (transcriptomics and proteomics) approach, or by integrating epigenetics (miRNA or lncRNA) with transcriptome data. Often, the analyses beyond transcriptome profiling are targeted analyses, conducted to replicate the initial RNA findings, and/or to propose a potential mechanism by which the transcriptome could have been altered.

Probably one of the most ambitious efforts to assess the opportunities and challenges in using integrated multi-omics data for health assessment and prediction comes from the integrated personal omics profiling (iPOP) project [81]. Initially proposed and started by longitudinally profiling a single individual over time with a broad set of omics approaches and clinical tests [82], the project has expanded and explored both the impact of lifestyle changes on long-term health, such as a brief weight gain followed by a weight loss [83], as well as longitudinal profiling of a diverse cohort to assess parameters that are useful in long-term health management and decision making [84]. The latter study followed 109 individuals at risk for type 2 diabetes on average for about 3 years, and used clinical assessments, wearable devices, and multi-omics profiling to derive predictive models for long-term health outcomes. The analysis highlights how the resulting data allow predictions and treatment decisions across a wide range of clinical specialties, including cardiovascular health, infectious diseases, and oncology. While this study has limitations and does not necessarily provide a clear blueprint for multi-omics studies and clinical applications, it is the first of its kind to assess the potential role of multi-omics data in health outcome predictions in an integrated manner.

Overall, this short summary highlights that the concept of precision medicine is more than the use of genetic variant information to inform clinical practice and the fact that the use multi-omics approaches to achieve precision medicine in the clinic is extremely valuable but still in its infancy. Not only in precision oncology, but in a wide range of human disease-related research, we are just beginning to generate comprehensive, unbiased data that are truly multi-omic. Only with the generation of these different omic datasets from the same biological samples will we be able to develop the necessary analytical (statistical) and annotation tools that will help us interpret these complex datasets, and help extract biological and clinically relevant information. The Trans-Omics for Precision Medicine (TOPMED) Initiative of the NHLBI is one example of a large-scale effort that is beginning to generate more comprehensive omics data for a wide range of study cohorts, based on already existing studies and sample material. Generating multi-omics data is expensive and time-consuming, and the usefulness of the resulting data critically depends on the availability of suitable tissue samples and biopsy material collected in a way that allows the effective analysis of tissue transcriptome, epigenome, proteome, and metabolome. Until recently, biopsy samples were mainly collected for transcriptome analysis or mutation discovery, and accordingly, samples were often collected in buffers that stabilized RNA and DNA, but essentially render the sample material unsuitable for protein or metabolite profiling. Future studies will need to address this so that we can generate the necessary comprehensive multi-omics data and begin to untangle the complex biological mechanisms that control tumor formation and progression, the development of resistance to treatment, and the risk of recurrence. We will need to focus on the characterization of integrated multi-omics signatures and mechanisms rather than small sets of putative diagnostic biomarkers that are currently being marketed to clinicians if we hope to truly advance precision medicine and precision oncology using multi-omics approaches.

## Figures and Tables

**Figure 1 ijms-20-04781-f001:**
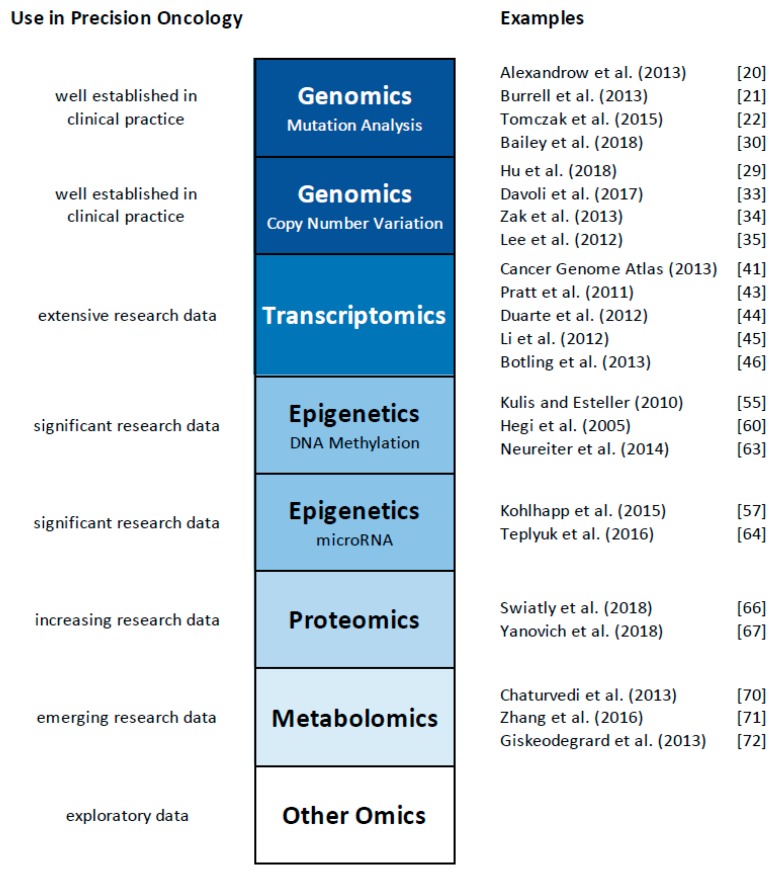
Summary of the applications of individual omics technologies to study cancer and other human disorders.

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
