# Peer review of "The Need for Multi-Omics Biomarker Signatures in Precision Medicine"

_ijms, 2019, doi:10.3390/ijms20194781_

Round 1

Reviewer 1 Report

This manuscript by Olivier et al. discussed a rather important topic and definitely worth publishing. The authors discussed two approaches to investigate multi-omics data: de novo approach and prior knowledge-based approach. However, some major modifications should be made.

1. For such a complex topic, the authors should make at least one figure to illustrate their ideas. So the readers can easily follow. In such a big and ever-evolving topic, at least one figure about the challenges of the multi-omics field should be made.

2. Lines 49-57 in the Introduction should be moved to the 'Challenges and Opportunities‘ part.

3. In lines 125-132, the authors discussed the usage of NGS for studying chromosomal abnormalities, I'd like to suggest them to discuss the latest studies such as inter-chromosomal recombination (IHR)-caused gene conversion using NGS (pubmed id: 30134973).

Author Response

Thank you for the review comments, we have addressed the issues raised as follows:

We are now including a figure that should help highlight the different omics approaches, their state relative to clinical impact, and key references, and make it easier for the reader to follow. We have carefully discussed the request to move this question, but feel it is important to set the stage for this problem already in the introduction.  The same points are being raised in the end of the manuscript, so despite the recommendation, we feel that moving this section would weaken the introduction, and not highlight this key challenge. We have added a brief discussion, as suggested. Thank you for pointing out this additional application for WGS data in oncology!

Reviewer 2 Report

The authors summarized the omics technologies in recent progress. Because few concrete examples were discussed and any figure/table reflecting the attractive technologies was not submitted, it is difficult to be well understood by readers, and be cited by numbers of other scientists’ papers.

This reviewer agrees with the present topic for the review article, but more attractive manuscript should be welcome for further consideration.

Author Response

We would like to thank the reviewer for the comments. We have now added a figure to the manuscript that should facilitate the concepts and technologies discussed in the paper. In addition, we have now included a brief discussion of recent true multi-omics studies (lines 327-340) to serve as an example for the goals and opportunities of multi-omics studies. We hope that this will help make the manuscript more attactive.

Round 2

Reviewer 1 Report

The authors have addressed all my concerns.

Reviewer 2 Report

Interesting figure has been provided and the revised statement had been added. This reviewer feels to agree their revisions.

However, some table(s) with raw data and/or some gene names/gene product names extracted through each level (mRNA, DNA, protein, etc) for each endpoint (diagnosis, molecular mechanisms of disease, toxicology, pharmacology, injury, etc) may be summarized as success examples of omics technology. only short sentences had been added to explain some results including the authors' work. these additional tables (or figures) may greatly help to understand by readers with potential higher citation affecting future work.